# The association between local hospital segregation and hospital quality for medicare enrollees

Ellesse-Roselee L. Akré[1,2]*, Deanna Chyn[2], Heather A. Carlos[3], Amber E. Barnato[2], Jonathan Skinner[2,4]

1 Bloomberg School of Public Health, Johns Hopkins University, Baltimore, Maryland, United States of America, 2 Geisel School of Medicine at Dartmouth, The Dartmouth Institute for Health Policy, and Clinical Practice, Hanover, New Hampshire, United States of America, 3 Dartmouth Cancer Center, Lebanon, New Hampshire, United States of America, 4 Department of Economics, Dartmouth College, Hanover, New Hampshire, United States of America

* eakre1@jhu.edu

## Abstract

There is considerable racial segregation in U.S. hospitals that residence alone cannot explain. Little is known about how this patient sorting affects racial inequalities. We use 2019 Medicare claims data linked to CMS Overall Hospital Quality Star Ratings to measure how the sorting of Black patients to different hospitals within the same healthcare markets affects racial gaps in hospital quality. Defining a hospital's market based on driving time, we use the local hospital segregation (LHS) index to measure racial sorting within a market; this measures the hospital's disproportionate share of Black patients relative to the hospital's market. In the sample of 2,163 hospitals, we find a 10 percent point increase in the LHS was associated with a 79% increase in the risk of being admitted to a lower-quality hospital. Thus, hospitals receiving a disproportionate share of Black patients due to market segregation are of systematically lower quality. A better understanding of hospital choice drivers could help reduce racial inequalities in health outcomes.

## Introduction

Racial disparities in access to quality healthcare among Medicare beneficiaries remain a persistent challenge in the United States. Extensive research has documented significant differences in healthcare access, utilization, and outcomes between Black and White individuals [1–6]. Studies on the Medicare population have demonstrated that Black and White Medicare patients are treated by different physicians, with Black patients being more likely to have physicians who are not board-certified and who report being unable to provide high-quality care to their patients [7]. Black Medicare patients are more likely to be admitted to a high-mortality hospital than White Medicare patients, even when they live closer to a low-mortality

**Data availability statement:** Data is not shared publicly but is available through a data use agreement with the Centers for Medicare and Medicaid Services. Send inquiries to datauseagreement@cms.hhs.gov.

**Funding:** National Institute on Aging (P01AG019783) and GeoSpatial Resource, a section of the Biostatistical and Bioinformatics Shared Resource at the Dartmouth Cancer Center with NCI Cancer Center Support Grant 5P30CA023108. The funders had no role in study design, data collection and analysis, decision to publish, or preparation of the manuscript.

**Competing interests:** The authors have declared that no competing interests exist.

hospital [4]. Some studies attribute these inequalities to geographic variation in clinical practices and residential segregation [1,8]. However, a recent study demonstrated that when controlling for the racial composition of the surrounding area, nearly 80 percent of acute care hospitals in the United States that treat Medicare fee-for-service patients did not admit a representative share of Black beneficiaries relative to their market areas [9].

Prior studies, such as Dimick et al., examined how residential segregation and proximity influence Black patients' likelihood of receiving care at low-quality hospitals [10]. However, these approaches do not capture the systematic sorting of Black patients into lower-quality hospitals across the full range of admissions and across all available options in their immediate markets. Beyond residential segregation, segregation within the health care system itself shapes access. Recent work mapping patient-sharing physician networks shows that Black–White network segregation is positively correlated with residential segregation yet lower in magnitude, and it differentially influences access to the best-available hospitals for CABG by market context [11]. The mechanisms by which local patterns of hospital admissions contribute to these disparities have not been fully explored. This study extends the literature by leveraging the Local Hospital Segregation (LHS) index, a novel a within-market measure of hospital sorting that captures the realized distribution of admissions across proximate hospitals [9]. By applying this index to a national Medicare population and linking it to the comprehensive, multidimensional hospital quality metrics of the CMS Star Rating, we offer new evidence of how within-market sorting beyond neighborhood residence drives inequities in the quality of care received by Black beneficiaries.

We relied on the Medicare fee-for-service population aged 65 and older because everyone in that population receives insurance coverage and can choose any U.S. Medicare-certified hospital. We hypothesized that Black patients were more likely to be admitted to a lower-quality hospital than to other hospitals nearby. Using regression analysis at the hospital level, we estimated the association between hospital quality and the LHS while controlling for the degree of residential segregation, hospital ownership, Census region, receipt of disproportionate share payments, market share, and teaching status. In sensitivity analysis, we considered the association between the LHS and the 7 individual quality components of the overall Medicare star rating, as well as stratifying by the size of the market, and by emergent and non-emergent admissions.

## Materials and methods

The project was approved by the Dartmouth Committee for the Protection of Human Subjects. The requirement for securing informed consent was waived because the data were deidentified and not collected as part of the study. The Centers for Medicare and Medicaid Services (CMS) issued a data use agreement (DUA) for the project. We followed the Strengthening the Reporting of Observational Studies in Epidemiology (STROBE) reporting guidelines. All statistical analyses were performed using STATA 18.0. This data analysis was conducted between November 1, 2022 and August 8, 2025.

## CMS Quality Star Rating

Our primary outcome measure was the 2020 Medicare Overall Quality Star Rating (based mainly on 2019 response data). The overall rating was based on seven weighted groups (or "group scores"): mortality, safety, readmission, patient experience, efficient use of medical imaging (efficiency), timeliness of care (timeliness), and effectiveness of care (effective); see S1 Table. The group scores have been linked to national quality initiatives such as the CMS Hospital Value-Based Purchasing program [12]. Each group score was standardized so that a higher score indicated a better outcome; thus, the directions for mortality and readmission scores were changed so that lower mortality or readmission rates received a higher score and, hence, a better star rating. These seven sections were weighted and combined into one composite score, the Overall Star Rating [13]. If a hospital was missing data for one or more groups, we reproportioned the weights of the missing groups and distributed them across the present groups. Previous studies demonstrated that favorable overall CMS Star Ratings were associated with better patient experiences and clinical outcomes [14–16].

## Study sample

The hospitals included in the study were acute care or critical access hospitals (excluding emergency hospitals) located within the contiguous United States. We identified 4,640 acute care or critical access hospitals providing treatment to any 2019 Medicare fee-for-service beneficiaries across 48 states (for a total of 11,304,184 admissions); our initial sample is the 8,381,731 hospital admissions with valid geospatial information within 30 minutes of the admitting hospital. We excluded 211 hospitals lacking the necessary geospatial data to construct the market area, resulting in 4,429 hospitals. In compliance with CMS reporting requirements, we excluded hospitals with fewer than 11 Black patients with eligible hospitalizations (which led to excluding 2,063 hospitals), fewer than 11 patients not identifying as Black (5 hospitals), and with fewer than 11 patients for both groups (76 hospitals), leading to 2,285 hospitals. For statistical precision, we removed 42 additional hospitals with fewer than 100 total hospitalizations of area Medicare patients aged 65 years or older, resulting in 2,243 hospitals. As a final step, we excluded 80 hospitals without a 2020 Star Rating. Thus, our exclusion rules resulted in slightly less than half of the original hospital count [2,163], with 12.8% fewer admissions than the initial sample (7,309,863 in total). Further details on the selection process are provided in S1 Fig.

## Exposure variables

Our primary exposure measure was the local hospital segregation (**LHS**) index, which was constructed using the 2019 Medicare Master Beneficiary Summary File and 2019 MedPAR data; see Akré et al. for further discussion [9]. The measure captures the difference between the fraction of inpatient Black admissions at a given hospital and the corresponding fraction of Black admissions in its market area, defined as ZIP codes within a 30-minute drive of the hospital. A hospital value of −0.10, for example, means that the fraction of Black hospital admissions in that hospital is 10 percentage points lower than that in the market area.

The secondary exposure measure was the fraction of Black admissions in the market area. Hospitals excluded from the analytic sample were not included in the denominator when estimating the market-level Black admission share for the LHS index. The actual share of Black admissions in a given hospital is equal to the LHS plus the market-level fraction of Black admissions; thus, it is a decomposition by which we can identify separately the association between hospital star ratings and (a) the market-level degree of segregation and (b) the within-market segregation as measured by the LHS. The market-level fraction of Black admissions is common to all hospitals in that market.

## Covariates

Using the 2019 American Hospital Association Survey and Medicare claims data, our fully adjusted models included the hospital Census region (Midwest, Northeast, South, and West), market share (the number of hospitals in a hospital's

market area), ownership (for profit/nonprofit), teaching hospital designation (teaching hospital/non-teaching hospital), and receipt of payment as a disproportionate share hospital (DSH) (DSH hospital/non-DSH hospital). These variables were included as control variables in the logistic and linear regressions because they have been associated with quality of care [17,18] and the LHS [9].

## Statistical analysis

We first considered the distribution of hospital Star Ratings for three groups: hospitals with an LHS significantly below 0, those with an LHS significantly greater than 0, and those with an LHS not significantly different from zero. We then used the entire sample to estimate the probability of being admitted to a low-quality hospital (defined as 1 or 2 stars) with a 0.10 change in the LHS, adjusting for the fraction of Black admissions in the hospital's market area. We used a logistic regression model and converted all estimates to probabilities using the Margins command in STATA.

We also considered a fully adjusted model including the LHS and Black admissions variables along with the covariates noted previously. In addition, we considered the association between the LHS and the seven components of the overall Star Rating; these measures were transformed into z-scores (e.g., the standard deviation is one) for the entire sample of hospitals.

For sensitivity analysis, we also consider two generalizations of the basic model. First, we hypothesize that in smaller markets with the highest concentration – for example those with just 2 hospitals –we would observe the greatest degree of sorting and adverse effects on hospital quality; thus, we stratified our analysis by the size of the market: small (1–2 hospitals), medium (3–6 hospitals), and large (7 + hospitals). Second, we stratify hospitals into quintiles of their LHS score to test whether those in the highest quintile are disproportionately more likely to be low quality (one or two star) (see S2 Table).

We also considered 3 additional robustness tests. The first tested the sensitivity of our findings to the admission type; for this reason, we stratified the models by emergent and non-emergent admissions (i.e., urgent and elective). The second used all hospital rankings in an ordered logistics model to estimate the association between the LHS and the probability of being admitted to a 1-, 2-, 3-, 4- or 5-Star hospital. Finally, we changed the market definition to a 15-minute driving time rather than 30-minute to better reflect potential travel barriers such as infrequent public transportation.

There were several limitations associated with this study. First, the data were cross-sectional and claims data may not reflect differences in underlying health status. However, the CMS claims data were used solely to measure patient flows (conditional on hospital admission); risk-adjustment issues were of greater concern for the CMS quality measures. We acknowledge that our patient assignment is limited to inpatient hospital admissions, and that the patterns of outpatient visits in hospitals (or associated medical practices) may differ.

Second, several hospitals had missing data in group scores but were still assigned an overall star rating. Previous studies demonstrated limitations in the validity of cross-group comparisons between hospitals with varying completeness of data [19]. Third, our measure of hospital quality was the average quality across all hospital patients and did not capture differential quality between Black patients and patients not self-identifying as Black. Most evidence on Black-White differences in hospital care quality suggests that the variation is primarily across hospitals; [5] less is known about within-hospital differences in quality between Black and White patients. By restricting our analyses to only patients admitted to the hospital (and not the population of Medicare fee-for-service beneficiaries living in the region), we sought to adjust for health care "need." Nevertheless, it is still possible that Black and other patients were admitted for systematically different reasons, although our sensitivity analysis found no difference between emergent and non-emergent admissions.

Fourth, our analysis was limited to the fee-for-service Medicare population. While the fraction of Medicare Advantage enrollees is growing over time, we did not use these enrollees because they are often subject to narrow networks, which would restrict the choice of hospitals within their markets, thus biasing our results.

A final limitation is the possibility of bias in the hospital star rating scores because of inadequate risk adjustment. While the lack of association between the LHS and the mortality score does not point towards bias in the measurement of mortality, we note that other measures, such as readmission rates or safety, may exhibit a larger degree of bias.

## Results

Table 1 shows that since hospitals with positive LHS values tend to balance those with negative values within a market, the mean LHS approximated zero (mean = 0.42, SD = 10.28). Thirty-five percent (N = 753) of hospitals received a 1- or 2-Star rating. Because 1–2 Star hospitals tend to be lower volume, they collectively account for a much smaller share of admissions than their share of hospitals would suggest, with only about 10 percent of patients admitted to 1–2 Star hospitals. Hospitals with a 1- or 2-Star rating treated a larger share of Black patients than hospitals with 3 or more stars (19.1% versus 12.6% of hospitalizations, respectively); they were more likely to be private for-profit (27.1% versus 15.9%); and they were more likely to be teaching hospitals (62.3% versus 45.8%). On average, hospitals with a 1- or 2-Star rating exhibited an overrepresentation of Black admissions relative to the market racial composition (LHS = 3.03, S.D., 12.30). In contrast, hospitals with 3 or more Stars exhibited an underrepresentation of Black admissions (LHS = −0.97 (S.D., 8.71)).

**Table 1. Characteristics of hospitals treating ever-hospitalized medicare fee-for-service enrollees aged at least 65 years, 2019.**

| Characteristic | All hospitals (N = 2,163) | | 1-to 2-Star (N = 753) | | 3- to 5-Star (N = 1,410) | |
|---|---|---|---|---|---|---|
| | Mean | S.D. | Mean | S.D. | Mean | S.D |
| Local Hospital Segregation index, mean (S.D.), avg % | 0.42 | 10.28 | 3.03 | 12.30 | −0.97 | 8.71 |
| Hospitalizations of Black patients at the profiled hospital | | | | | | |
| Average number | 446.83 | 693.39 | 586.59 | 826.40 | 372.19 | 597.69 |
| Average percentage | 14.83 | 16.53 | 19.09 | 20.10 | 12.55 | 13.73 |
| Hospitalizations of Black patients in the hospital's market area | | | | | | |
| Average number | 5184.31 | 7951.89 | 6954.39 | 10128.95 | 4239.01 | 6300.82 |
| Average percentage | 14.41 | 11.87 | 16.06 | 13.12 | 13.52 | 11.06 |
| | No. | % | No. | % | No. | % |
| Disproportionate Share Hospital (DSH) | 1906 | 88.12 | 717 | 95.22 | 1189 | 84.33 |
| Census region | | | | | | |
| Northeast | 331 | 15.30 | 156 | 20.72 | 175 | 12.41 |
| Midwest | 414 | 19.14 | 101 | 13.41 | 313 | 22.20 |
| South | 1056 | 48.82 | 376 | 49.93 | 680 | 48.23 |
| West | 362 | 16.74 | 120 | 15.94 | 242 | 17.16 |
| Hospitals in market[a] | | | | | | |
| 1-2 | 417 | 19.28 | 141 | 18.73 | 276 | 19.57 |
| 3-6 | 740 | 34.21 | 253 | 33.60 | 487 | 34.54 |
| 7+ | 1006 | 46.51 | 359 | 47.68 | 647 | 45.89 |
| Hospital ownership | | | | | | |
| Private/Church (nonprofit) | 1209 | 55.89 | 357 | 47.41 | 852 | 60.43 |
| Physician/other | 208 | 9.62 | 68 | 9.03 | 140 | 9.93 |
| Private (for profit) | 428 | 19.79 | 204 | 27.09 | 224 | 15.89 |
| Government | 318 | 14.70 | 124 | 16.47 | 194 | 13.76 |
| Teaching hospital[b] | 1115 | 51.55 | 469 | 62.28 | 646 | 45.82 |

[a]A hospital's market area is defined as ZIP codes whose centroids are within 30-minute driving distance of the hospital's geocoded address. It includes the profiled hospital (i.e., no market is smaller than 1). S.D. denotes standard deviation.

[b]Five hospitals had unknown status.

 

Fig 1 shows the full distribution of Star Ratings for three categories of hospitals: those with significant negative LHS values (underrepresentation of Black patient admissions relative to the market area), those with LHS values not significantly different from zero, and those with significant positive LHS values (overrepresentation of Black admissions relative to the market area). There was a clear shift in the distribution of hospital Star Ratings, with consistently lower ratings for positive-LHS hospitals (Panel 3 versus Panel 1).

Table 2 presents the primary logistic regression results. The coefficients (converted from odds ratios to probabilities) for the LHS in the minimally adjusted model (0.079, 95% CI 0.060–0.099) imply that a 0.10 change in the LHS was

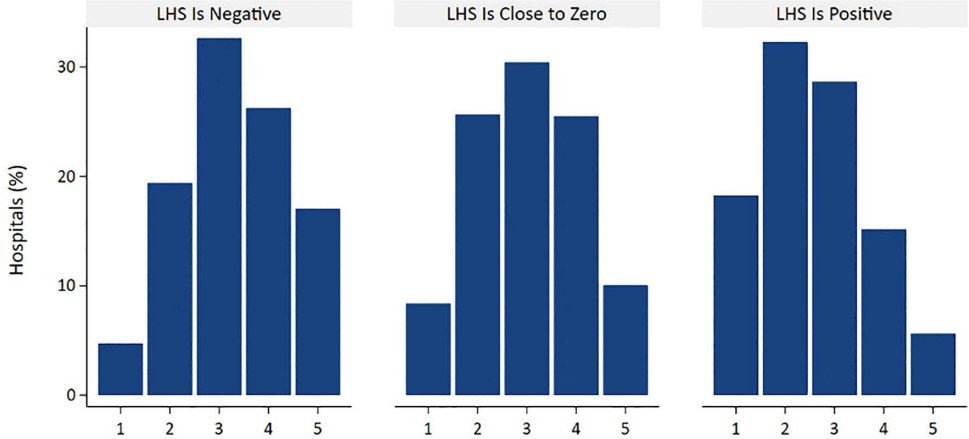

**Fig 1. Distribution of hospital Star Rating by local hospital segregation index.**

**Table 2. Association between hospital quality (1- or 2-star) and hospital characteristics using CMS data from 2019.**

| | Model 1 | | | Model 2 | | |
|---|---|---|---|---|---|---|
| | N Hosps. | LHS (10 pct pts) | Market admits[a] (10 pct pts) | N Hosps. | LHS (10 pct pts) | Market admits* (10 pct pts) |
| All hospitalizations | 2,163 | 0.079 (.060,.099) | 0.030 (.014,.047) | 2,158 | 0.060 (.041,.079) | 0.041 (.023,.058) |
| *Stratified by the Number of Hospitals in the Market* | | | | | | |
| Small markets (1–2 hospitals) | 417 | 0.046 (−.052,.145) | 0.013 (−.020,.047) | 417 | 0.010 (−.086,.107) | 0.013 (−.023,.049) |
| Medium markets (3–6 hospitals) | 740 | 0.089 (.031,.147) | 0.031 (.0003,.062) | 738 | 0.067 (.009,.124) | 0.031 (−.004,.066) |
| Large markets (7+ hospitals) | 1,006 | 0.076 (.055,.096) | 0.043 (.019,.068) | 1,003 | 0.058 (.037,.078) | 0.065 (.039,.092) |
| *Stratified by Emergency and Non-Emergency Hospitalizations* | | | | | | |
| Emergency hospitalizations | 2,160 | 0.066 (.049,.084) | 0.031 (.015,.046) | 2,155 | 0.050 (.033,.068) | 0.040 (.023,.057) |
| Non-emergency hospitalizations | 2,124 | 0.078 (.056,.099) | 0.036 (.014,.058) | 2,119 | 0.058 (.037,.078) | 0.051 (.028,.074) |

**Note**. Values in the table are marginal effects with confidence intervals in parentheses. Model 1 = adjusted for LHS and % Black residents from the market admitted to any hospital. Model 2 = additionally adjusted for market share, ownership, teaching status, disproportionate share hospital (DSH) status, and census region.

[a]Market admits = the percentage of hospital admissions in a hospital's market area of Black patients.

associated with a 7.9 percentage point (or a 79%) increase in the likelihood of being admitted to a 1- or 2-Star hospital; the fully adjusted model yielded a slightly smaller coefficient (0.060, 95% CI 0.041–0.079). By contrast, the association between the percentage of Black hospital admissions in the market (a measure of residential segregation) and the probability of being admitted to a 1- or 2-Star hospital was positive but of a smaller magnitude; 0.030 (95% CI 0.014–0.047) for the minimally adjusted model and 0.041 (95% CI, 0.023–0.058) for the fully adjusted model.

We also present results stratified by the degree of concentration in the market. The next three rows present results for Model 1 and Model 2, stratified by the size of the market. For the small market, with 1−2 hospitals, the coefficient in Model 1 on LHS is 0.046, and not significantly different from zero; in medium-sized markets with 3−6 hospitals, the coefficient is 0.089 (95% CI, 0.031–0.147), and for the large-sized markets (7+) the coefficient is 0.076 (95% CI, 0.055–0.096); the pattern of coefficients is similar, but attenuated for Model 2. That is, the association between LHS and our quality measures is the strongest among the large and especially medium-sized markets, suggesting a greater degree of segmentation within larger markets (see Table 2). This indicates that the most pronounced disparities occur in larger and medium-sized markets where the larger number of hospitals appears to contribute a greater degree of patient sorting. Finally, Table 2 shows that splitting out emergency and non-emergency hospitalizations yielded similar results.

Fig 2 considers the full set of hospitals ranked from 1 to 5 Stars using an ordered logistics regression, presented just for the fully adjusted model (all results reported in S2 Table). The association between the LHS and the probability of being admitted to a 1- or 2-Star hospital was significantly positive, unassociated for 3-Star hospitals, and significantly negative for 4- or 5-Star hospitals.

In Appendix S3 Table, we provide the estimates of both Model 1 and Model 2 but with hospitals stratified into quintiles where the lowest quintile exhibits a negative LHS and the highest quintile measures hospitals with the highest LHS. There is a positive and nonlinear association between the LHS and the likelihood of being admitted to a low quality (1 or 2 star) hospital, but with diminishing effects of an increase in LHS scores; indeed, for the fully adjusted model (Column 3) there is no difference in moving from the 4th to the 5th quintile of the LHS. That is, the hospitals in locally segregated markets with a large (in magnitude) negative LHS exhibit the lowest likelihood of being 1- or 2-Star hospitals.

We also considered the association between the LHS and each of the seven group scores; results are in the S4 Table and S2 Fig. There was a negative association between the LHS and the group score z-scores for six of the seven group scores. The exception was for the mortality subgroup, where coefficients for the minimally adjusted LHS (and the

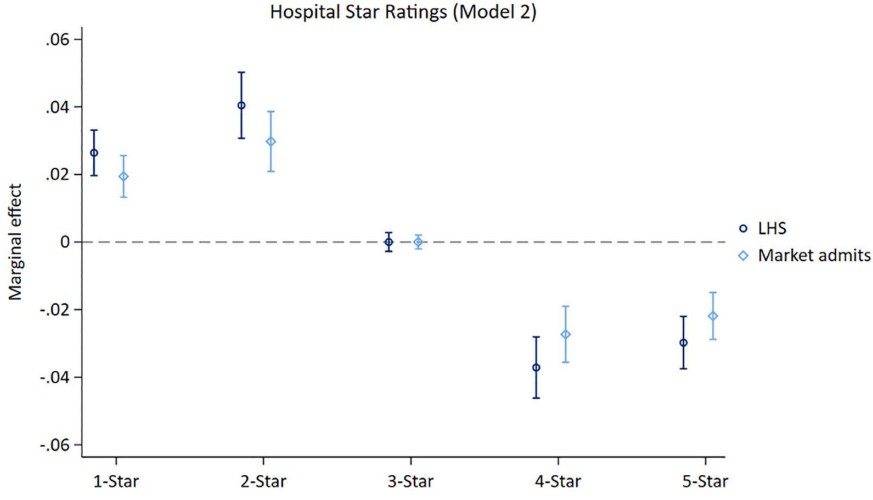

**Fig 2. Fully adjusted ordered logistics regression for 1 through 5 CMS quality.**

percentage Black admissions in the market for both models) were positive (e.g., lower mortality) and approached statistical significance. Finally, in an analysis in S5 Table, we find similar results when we use a 15-minute driving radius for our definition of markets.

## Discussion

There is long-standing literature on the association between residential segregation and healthcare quality [1,6,8]. In this paper, we contrast residential segregation (as measured by the total market share of Black hospital admissions) with a different source of inequality: the differential sorting of Black patients to hospitals within the same geographic market. As expected, we found that increased residential segregation increased the likelihood of patients being hospitalized at a lower-quality hospital (as have other studies). Our most notable finding was that patient sorting *within* local area hospitals contributed to an even greater likelihood of being admitted to a lower-quality hospital. Further, our findings demonstrated that patient sorting was associated with decreased safety, patient experience, effectiveness, timeliness, and increased readmission, all critical components of patient quality.

To our knowledge, our findings are the first to demonstrate that when Black patients are near several hospitals, they are more likely to be admitted to a lower CMS Star-Rated hospital. The study findings are consistent with previous evidence that Black patients encounter less patient safety and worse patient experiences [20]. Indeed, because we did not capture differences in quality by race within hospitals, we likely underestimated the frequency with which Black patients receive lower quality [3]. While we acknowledge the possibility of statistical biases in the CMS star ranking, the fact that we found no difference in mortality for hospitals with a higher LHS suggests that the other quality subcategories are reflective of true quality gaps.

We acknowledge there are likely reasons why patients bypass a higher-quality hospital to seek care. There may be issues of availability or accessibility that have not been captured in any study to date. Yet, if patient choices are induced by unfriendly or hostile reception by staff at the closer and higher-quality hospitals or by perceptions that the hospital is untrustworthy, it is a fundamental barrier to health equity. Further qualitative research could play a role in identifying what factors, such as intuitional racism and "discriminatory policies and practices carried out…[within and between individual] state or non-state institutions [21]" are most salient in influencing patient decisions.

Previous studies on patient hospital selection factors included (a) physician and employee behavior and the clinic environment [22,23], (b) the reputation of the hospital [22,23], (c) network competition and the hospital's market competition, location, and accessibility [24–27], and (d) the variety of services [26,28]. These studies provided insight into what may influence patient choice in hospitals but were not stratified to understand patient choice by race or ethnicity. According to frameworks for patient-centered shared decision-making [29], the decision would need to be fully informed and values-aligned but not related to differential feelings of inclusion, belonging, or trustworthiness by Black patients.

Our findings demonstrate that while information gaps regarding hospital quality may contribute to observed patterns of hospital choice, they are insufficient to fully explain the persistent sorting of Black patients into lower-quality hospitals. Our analyses, focused on Medicare fee-for-service beneficiaries who have broad nominal choice among hospitals, show that disparities persist even when several hospitals are located within close geographic proximity and quality information is publicly available [9]. This is consistent with prior research showing that, although all patients—including Black patients—use social networks and trust in their communities during decision-making, these behaviors alone do not account for the magnitude of observed segregation in hospital use [7,8]. Rather, the literature and our own results indicate that structural, economic, and institutional factors—such as insurance networks, hospital referral patterns, historical relationships, and experiences of inclusion or exclusion within health care settings—play a predominant role in shaping hospital selection for Black patients [8,9,30]. Our results, particularly for emergency care where patient choice is less likely to be important, reinforce the conclusion that supply-side constraints and systemic barriers, rather than information (and demand more generally), underpin the persistent inequities in hospital quality experienced by Black communities [30,31].

Complementing our focus on within-market hospital sorting, Landon et al. also show that primary care physicians share Black patients with fewer specialists than White patients across multiple markets—even after equalizing patient volumes—indicating race-linked referral network fragmentation that likely contributes to downstream differences in hospital access and quality [32].

In both the minimally and fully adjusted models, the patient mortality score (which is higher when mortality is lower) was positively associated with higher LHS index scores, although neither result was significant. This lack of a strong association was consistent with Pandey et al. (2020), who found very modest differences by both race and hospital quality in 30-day mortality following acute myocardial infarction (AMI) [33]. This finding may be due to the likelihood that hospitals with a positive LHS are teaching hospitals, which often exhibit better mortality outcomes [17].

The current study makes several significant contributions beyond prior work in this field. First, by employing the LHS index, we provide a direct and scalable approach to assess differential patterns of hospital admission within local markets, distinct from traditional measures based solely on residential segregation or broad regional patterns. Second, our analysis demonstrates that Black patients are more likely to be admitted to lower-quality hospitals even when higher-quality options are geographically proximate, and network access is nominally equal—implicating within-market institutional and structural factors that perpetuate care disparities. Third, by leveraging comprehensive hospital quality ratings, our findings reveal that local hospital segregation is not linked to mortality but is related to the remaining multiple domains of patient safety, experience, efficiency, and timeliness. Together, these findings shift the focus from geographic and patient-side explanations alone toward the recognition of supply-side and systemic barriers rooted in how hospitals admit and serve patients within their markets. This broader analysis thus may help to design policy interventions aimed at advancing equity in U.S. health care delivery.

## Conclusion

In sum, local hospital segregation (LHS) was demonstrated to have a negative relationship with hospital quality. The over-representation of Black patients in a hospital (relative to its potential market share of Black admissions) was associated with worse safety, patient experience, timeliness, effectiveness, efficiency, and higher readmissions (but not worse mortality). While the healthcare system is unlikely to affect residential segregation, it has the scope to influence patient decisions and hospital policies, reducing the degree of patient sorting within markets and thereby reducing inequities in healthcare quality for Black hospital patients.

## Supporting information

**S1 Fig. Sample construction.**
(DOCX)

**S2 Fig. Hospital Star Standardized Group Scores.**
(DOCX)

**S1 Table. CMS Overall Hospital Quality Star Rating (2020) group score definitions.**
(DOCX)

**S2 Table. Association between hospital quality (Star rating) and hospital characteristics using CMS data from 2019: Ordered logit results.**
(DOCX)

**S3 Table. Association between the probability of being admitted to a low-quality hospital (1- or 2-Star) and the Local Hospital Segregation Index: Logistics results.**
(DOCX)

**S4 Table. Association between hospital quality and hospital characteristics: Logistics (1- or 2-Star) and OLS (group scores) results.**
(DOCX)

**S5 Table. Association between hospital quality (1- or 2-Star rating) and hospital characteristics: Market definition using a 15-minute driving radius.**
(DOCX)

## Author contributions

**Conceptualization:** Ellesse-Roselee L. Akré, Jonathan Skinner.

**Data curation:** Ellesse-Roselee L. Akré, Deanna Chyn, Heather A. Carlos.

**Formal analysis:** Ellesse-Roselee L. Akré, Deanna Chyn.

**Funding acquisition:** Amber E. Barnato.

**Methodology:** Ellesse-Roselee L. Akré, Jonathan Skinner.

**Supervision:** Jonathan Skinner.

**Visualization:** Deanna Chyn.

**Writing – original draft:** Ellesse-Roselee L. Akré, Deanna Chyn, Jonathan Skinner.

**Writing – review & editing:** Ellesse-Roselee L. Akré, Heather A. Carlos, Jonathan Skinner.

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
