## [Decision Letter · Decision Letter 0]

2 Jul 2025

Dear Dr. Akre,

Thank you for submitting your manuscript to PLOS ONE. After careful consideration, we feel that it has merit but does not fully meet PLOS ONE’s publication criteria as it currently stands. Therefore, we invite you to submit a revised version of the manuscript that addresses the points raised during the review process.

I sent the paper to two referees and have now heard back from them. I’ve also had a chance to read your paper myself. This isn't an easy case - one referee suggested major revision and the other referee suggested reject. I love the idea you are pursuing with the paper and find it important.

The referees have offered a few suggestions to potentially assuage their concerns. On my read, I have to say that you need to add more details on the paper's contribution to literature (Referee 1). Also, I agree with Referee 2 that your paper would benefit from discussing more on mechanisms. Because of the importance of the question, I am willing to give you another round to try. I like the question you pursue and am willing to hear if you have a response to the concerns and thoughts raised by the referees.

We look forward to receiving your revised manuscript.

Sincerely,

Hansoo Ko, MD, PhD

Academic Editor

PLOS ONE

Journal Requirements: 

 [National Institute on Aging (P01AG019783) and GeoSpatial Resource, a section of the Biostatistical and Bioinformatics Shared Resource at the Dartmouth Cancer Center with NCI Cancer Center Support Grant 5P30CA023108]. 

[Support was provided by a diversity supplement award by the National Institute on Aging (P01AG019783) and GeoSpatial Resource, a section of the Biostatistical and Bioinformatics Shared Resource at the Dartmouth Cancer Center with NCI Cancer Center Support Grant 5P30CA023108]

  [National Institute on Aging (P01AG019783) and GeoSpatial Resource, a section of the Biostatistical and Bioinformatics Shared Resource at the Dartmouth Cancer Center with NCI Cancer Center Support Grant 5P30CA023108]. 

4. In the online submission form, you indicated that your data is available only on request from a third party. Please note that your Data Availability Statement is currently missing [the name of the third party contact or institution / contact details for the third party, such as an email address or a link to where data requests can be made]. Please update your statement with the missing information.

Reviewers' comments:

Reviewer's Responses to Questions

**Comments to the Author**

1. Is the manuscript technically sound, and do the data support the conclusions?

Reviewer #1: Yes

Reviewer #2: Partly

2. Has the statistical analysis been performed appropriately and rigorously?

Reviewer #1: Yes

Reviewer #2: No

3. Have the authors made all data underlying the findings in their manuscript fully available?

Reviewer #1: Yes

Reviewer #2: Yes

4. Is the manuscript presented in an intelligible fashion and written in standard English?

Reviewer #1: Yes

Reviewer #2: Yes

Reviewer #1: Major comments:

• My biggest concern is that the observed associations may be driven by markets with a small number of hospitals. LHS values are more likely to be extreme due to the “law of small numbers” where extreme outcomes are more common in small samples. In this study, roughly 20 percent of markets have only one or two hospitals, and about 50 percent have six or fewer. Since LHS values are likely to be more extreme in these small markets, the associations with hospital quality may be disproportionately driven by these cases. It would be valuable to repeat the analysis restricting to markets with at least 7 or 10 hospitals to assess the robustness of the findings in settings less susceptible to this small-sample variance artifact.

• The results are consistent with structural racial disparities in healthcare delivery and hospital quality, but the interpretation appears to lean toward a supply-side explanation. Implicitly, the findings assume that both Black and non-black patients both have full information about hospital quality. However, it may be the case that Black patients face greater information gaps regarding hospital quality, while non-Black patients have better access to such information. This alone could lead to differential sorting, even in the absence of discrimination or exclusion. In the presence of such information gaps, patients may rely on heuristics such as choosing the same hospitals as others in their community. This could reinforce patterns of segregation and sorting. Such behavior would still be consistent with the observed results but would reflect limited information or social learning rather than active exclusion. It would strengthen the paper to include a discussion of these potential demand-side mechanisms and to acknowledge that observed sorting may partly reflect differences in information, trust, or social networks.

Minor comments:

• The analytic sample drops from over 4,400 to 2,285 hospitals due to the exclusion of hospitals with fewer than 11 Black or 11 non-Black admissions. Given the size of this reduction, it would be helpful for the authors to report how many hospitals were excluded for each criterion separately (i.e., low Black vs. low non-Black counts). Please also

• It may also be useful to examine whether the associations between LHS and hospital quality are strongest in markets where the index hospital is the only one serving a meaningful number of Black patients. This could help clarify whether the observed quality disparities are concentrated in racially homogenous markets where all other hospitals are nearly “all-White.”

• Other clarifications:

o Clarify whether these excluded hospitals are included in the denominator when calculating the market-level Black admission share for the LHS index. I could not see it in the manuscript.

o The sentence stating that “a hospital value of -0.10… is 10% less than that in the market area” is misleading. Since the LHS measures an absolute difference, it should read “10 percentage points lower” to reflect the correct interpretation.

o In the regression models, the market-level fraction of Black admissions is included as a control alongside the LHS index. Please clarify whether this variable is calculated using the same method as in the LHS denominator, specifically whether the index hospital is excluded from this calculation.

Reviewer #2: This paper examines the association between Local Hospital Segregation (LHS) and Hospital Quality. Using 2020 hospital star ratings and 2019 patient flow from Medicare Claims to construct the LHS measure, the authors provide cross-sectional evidence that hospitals with a higher local hospital segregation index (i.e., admitting disproportionally more black patients) tend to be of lower quality (lower star rating). Although the topic is important and policy relevant, I do not think it contributes sufficient new knowledge to the literature. There has been ample evidence showing black patients are more likely to bypass nearby high-quality hospitals and receive care in poorer quality hospitals, see e.g., Dmick et al. (2013).

It would be more interesting if the authors could explore the potential mechanisms. For example, primary care physicians are patients’ entry points to the healthcare system. Is the reason black patients bypass high-quality hospitals because primary care physicians serving black communities tend to refer patients to specialists affiliated with poor-quality hospitals? If so, why?

The authors need to explain how they constructed the patient flow measure using Medicare claims data. Is it only based on the inpatient data, or both inpatient and outpatient data? What is the methodology? If we are only looking at hospital admissions based on inpatient claims, I would think the “hospital choice” is more determined by the referring physicians’ affiliation than the patients’ preferences. Can the authors explore outpatient choices as well?

Also, it is unclear if hospitals with high LHS are indeed of lower quality, or it is simply because CMS hospital star rating scores (not just the mortality scores) inadequately account for social risk factors. The paper mentions that their empirical findings reject the explanation of inadequate risk adjustment based on the insignificant association between mortality score and LHS, but this argument applies to the other group scores as well (e.g., readmission and safety).

Reference:

Dimick, Justin, et al. "Black patients more likely than whites to undergo surgery at low-quality hospitals in segregated regions." Health Affairs 32.6 (2013): 1046-1053.

**Do you want your identity to be public for this peer review?** For information about this choice, including consent withdrawal, please see our Privacy Policy

Reviewer #1: No

Reviewer #2: No

---

## [Author Response · Author response to Decision Letter 1]

12 Aug 2025

Response to reviewers is included in attached files.

---

## [Decision Letter · Decision Letter 1]

6 Oct 2025

Dear Dr. Akre,

Thank you for submitting your manuscript to PLOS ONE. After careful consideration, we feel that it has merit but does not fully meet PLOS ONE’s publication criteria as it currently stands. Therefore, we invite you to submit a revised version of the manuscript that addresses the points raised during the review process.

We look forward to receiving your revised manuscript.

Kind regards,

Hansoo Ko

Academic Editor

PLOS ONE

Journal Requirements:

Additional Editor Comments:

I would appreciate if the authors provide more details on the result in Appendix Table A.3.b.

- I see that the authors hypothesized that hospitals with the highest LHS experience the sharpest increase in the probability of being lower-quality hospitals, and what I see in the table seem opposite to the hypothesis. Please add more discussion on this.

- The authors stated that "the hospitals in locally segregated markets with a large (in magnitude) negative LHS exhibit the lowest likelihood of being 1- or 2-Star hospitals". My interpretation of the results is that the relationship between the LHS quintile and the outcome is positive "and nonlinear" (or, there isn't much difference between "largely segregated" hospitals and "extremely segregated" hospitals?). If I'm not mistaken, please add more detailed interpretation to the manuscript.

- I think this model only takes into account the index hospital's segregation measure (whether or not this hospital is the only segregated one in the local market is unclear). Thus, Appendix Table A.3.b does not seem to be a proper response to the Reviewer #1's comment #4 ("It may also be useful to examine whether the associations between LHS and hospital quality are strongest in markets where the index hospital is the only one serving a meaningful number of Black patients").

Reviewer's Responses to Questions

**Comments to the Author**

Reviewer #2: All comments have been addressed

2. Is the manuscript technically sound, and do the data support the conclusions?

Reviewer #2: Yes

3. Has the statistical analysis been performed appropriately and rigorously?

Reviewer #2: Yes

4. Have the authors made all data underlying the findings in their manuscript fully available?

Reviewer #2: (No Response)

5. Is the manuscript presented in an intelligible fashion and written in standard English?

Reviewer #2: (No Response)

Reviewer #2: The authors have added discussions on potential mechanisms and contributions to the literature. It addressed my previous concerns and comments.

**Do you want your identity to be public for this peer review?** For information about this choice, including consent withdrawal, please see our Privacy Policy

Reviewer #2: No

---

## [Author Response · Author response to Decision Letter 2]

29 Oct 2025

Thank you for the opportunity to respond to the editors questions regarding our article. Please find our response below as well as attached to the submission:

Additional editor comments:

I would appreciate if the authors provide more details on the result in Appendix Table A.3.b.:

• I see that the authors hypothesized that hospitals with the highest LHS experience the sharpest increase in the probability of being lower-quality hospitals, and what I see in the table seem opposite to the hypothesis. Please add more discussion on this.

o Thank you for this comment and we apologize for the confusion. Rather than the dependent variable being the number of stars (so that a positive number is a good thing), the regression instead predicts whether the patient is admitted to a low-quality hospital (so that a positive coefficient is a bad thing). To increase the clarity of this table we change the title to “Association between the probability of being admitted to a low-quality hospital (1- or 2-Star) and the Local Hospital Segregation Index: Logistics results”

• The authors stated that "the hospitals in locally segregated markets with a large (in magnitude) negative LHS exhibit the lowest likelihood of being 1- or 2-Star hospitals". My interpretation of the results is that the relationship between the LHS quintile and the outcome is positive "and nonlinear" (or, there isn't much difference between "largely segregated" hospitals and "extremely segregated" hospitals?). If I'm not mistaken, please add more detailed interpretation to the manuscript.

o Thank you for this suggestion. The following language was added to manuscript to clarify the interpretation: There is a positive and nonlinear association between the LHS and the likelihood of being admitted to a low quality (1 or 2 star) hospital, but with diminishing effects of an increase in LHS scores; indeed for the fully adjusted model (Column 3) there is no difference in moving from the 4th to the 5th quintile of the LHS.

• I think this model only takes into account the index hospital's segregation measure (whether or not this hospital is the only segregated one in the local market is unclear). Thus, Appendix Table A.3.b does not seem to be a proper response to the Reviewer #1's comment #4 ("It may also be useful to examine whether the associations between LHS and hospital quality are strongest in markets where the index hospital is the only one serving a meaningful number of Black patients").

o An excellent point that we should have made clearer. The problem with single hospitals in a market is that – except for cross-market shifts – the LHS is identically zero, since the hospital is exactly representative of its market. For this reason, we defined “small markets” as being those with 1 or 2 hospitals; thus in this small market setting, one could imagine a “White” and a “Black” hospital, perhaps as a legacy of the time when there was explicit segregation. To address the reviewer’s comment in Table 2 of our manuscript, we conducted analyses stratified by market size. In these small markets, the association between LHS and hospital quality was not statistically significant (coefficient = 0.010, CI: -0.086 to 0.107 for fully adjusted model), whereas the association was stronger and significant in larger and especially medium-sized markets. This suggests that segmentation effects may be most pronounced when there are several hospitals for patient choice within a market, rather than in situations with only one or two hospitals. We added the following language to the results sections to clarify the meaning of the findings: This indicates that the most pronounced disparities occur in larger and medium-sized markets where the larger number of hospitals appears to contribute a greater degree of patient sorting.

---

## [Editor Report · Decision Letter 2]

11 Nov 2025

The Association Between Local Hospital Segregation and Hospital Quality for Medicare Enrollees

PONE-D-25-24400R2

Dear Dr. Akre,

We’re pleased to inform you that your manuscript has been judged scientifically suitable for publication and will be formally accepted for publication once it meets all outstanding technical requirements.

*Wishing you continued success and meaningful contributions to the field.*

Best regards,

Hansoo Ko, MD, PhD

Academic Editor

PLOS ONE

---

## [Editor Report · Acceptance letter]

PONE-D-25-24400R2

PLOS ONE

Dear Dr. Akre,

I'm pleased to inform you that your manuscript has been deemed suitable for publication in PLOS ONE. Congratulations! Your manuscript is now being handed over to our production team.

Kind regards,

on behalf of

Dr. Hansoo Ko

Academic Editor

PLOS ONE